# Revealing the structures of megadalton-scale DNA complexes with nucleotide resolution

Massimo Kube[1,3], Fabian Kohler[1,3], Elija Feigl[1,3], Baki Nagel-Yüksel [1], Elena M. Willner[1], Jonas J. Funke[1], Thomas Gerling[1], Pierre Stömmer[1], Maximilian N. Honemann[1], Thomas G. Martin[2], Sjors H. W. Scheres [2] & Hendrik Dietz [1✉]

The methods of DNA nanotechnology enable the rational design of custom shapes that self-assemble in solution from sets of DNA molecules. DNA origami, in which a long template DNA single strand is folded by many short DNA oligonucleotides, can be employed to make objects comprising hundreds of unique DNA strands and thousands of base pairs, thus in principle providing many degrees of freedom for modelling complex objects of defined 3D shapes and sizes. Here, we address the problem of accurate structural validation of DNA objects in solution with cryo-EM based methodologies. By taking into account structural fluctuations, we can determine structures with improved detail compared to previous work. To interpret the experimental cryo-EM maps, we present molecular-dynamics-based methods for building pseudo-atomic models in a semi-automated fashion. Among other features, our data allows discerning details such as helical grooves, single-strand versus double-strand crossovers, backbone phosphate positions, and single-strand breaks. Obtaining this higher level of detail is a step forward that now allows designers to inspect and refine their designs with base-pair level interventions.

[1] Physik Department, Technische Universität München, Garching, Germany. [2] MRC Laboratory of Molecular Biology, Cambridge, UK. [3] These authors contributed equally: Massimo Kube, Fabian Kohler, Elija Feigl. ✉email: dietz@tum.de

Programmable self-assembly with DNA is a route to nano-fabrication[1–8] with applications emerging in a variety of fields[9,10]. The self-assembly reactions of such objects can yield monodisperse products[11] and the underlying design concepts in principle enable specifying target structures with chemical accuracy on the level of single bases, within objects designed to contain several thousand bases. As in any other field aimed at creating items of technology, also the structures of objects built with the methods of DNA nanotechnology must be validated experimentally. However, the accuracy and the depth of the available structural data in DNA nanotechnology remains poor compared to the structural data that are routinely generated in other fields of study such as de novo protein design[12]. Here, we addressed the problem of accurate structural validation with cryo-EM-based methodologies for determining the structures of up to megadalton-scale DNA objects in solution, together with molecular-dynamics-based methods for building pseudoatomic models in a semiautomated fashion. Our methods yield structures that afford improved detail compared to our own previous work[13] and to those of others[8,14] (see Supplementary Table S1), and now allow discerning details such as helical grooves, single-strand versus double-strand crossovers, backbone phosphate positions, and nick sites. Access to data with such a level of detail enables performing iterative geometry refinements on the level of strands or individual base pairs, as we will show. We predict that this capability will enable the field to move toward more advanced functionalities that require the accurate relative positioning of functional groups, such as molecular recognition, proximity-enhanced templated synthesis, near-field photonic effects, or even enzyme-like catalysis. Likewise, DNA–template-assisted structural determination of proteins[14–16] presumably will benefit from these improvements in cryo-EM methodology for DNA origami.

We determined cryo-EM maps for a library of multilayer DNA objects in honeycomb and in square-lattice packing (Fig. 1 and Supplementary Figs. S1–26). The library of objects includes four brick-like multilayer DNA origami objects (Fig. 1a) with increasing internal floppiness, a barrel-like 126-helix bundle (Fig. 1b), two variants of a multidomain object called Twisttower (Fig. 1c), a design variant v2 of a previously reported object called the Pointer (Fig. 1d), a small 16-helix bundle (Fig. 1e), four variants of a hinged-beam-like object (Fig. 1f), two variants of a dumbbell-like object (Fig. 1g), and five variants of a six-helix tube featuring asymmetric markers at either end as reporters for twist and one ten-helix tube (Fig. 1h). We also attempted to solve structures of variants of single-layer DNA origami tiles in square-lattice design (Rothemund Rectangle[1]), but were unsuccessful due to excessive conformational heterogeneity (Supplementary Fig. S27). The micrographs showed that the original, uncorrected tile with a crossover density corresponding to a twist density of 10.67 base pairs per turn assumes wrapped-up-like shapes in solution. The high degree of flexibility and the wrapped-up shape is in accordance with previous findings from simulations, on-support atomic force microscopy (AFM), negative-stain electron microscopy (EM), and small-angle X-ray scattering (SAXS) data[17–20], and should be taken into account for in-solution applications. Exemplary cryo-EM micrographs and acquisition details for each object are given in Supplementary Figs. S1–S27 and in Supplementary Table S2, respectively.

## Results

### Global twist deformation in a square-lattice.
The object termed Twisttower (Fig. 1c) is a complex of four cuboid-like domains fused together. The cuboids feature $2 \times 2$, $4 \times 4$, $6 \times 6$, and $8 \times 8$ helices, respectively, in a quadratic cross-sectional arrangement (Fig. 2a). The Twisttower design allowed us to systematically study twist deformations, and how they may be removed, in a square-lattice packing context as a function of cross-sectional area. In the cryo-EM map that we determined, each cuboid domain exhibited an independent right-handed twist deformation around the helical axis of the domain (Fig. 2a). Such twist deformations are expected to arise a priori for square-lattice designs with default eight-base-pair crossover spacing because the design specifics create right-handed internal torques[4]. These torques are produced by helical underwinding from 10.5 to 10.67 base pairs per turn imposed by the square-lattice connectivity. The global twist observed in each Twisttower domain decreased with the increasing cross-sectional area (Fig. 2a, left). To remove the global twist deformation, the internal design specifics must be changed such that the right-handed internal torques are mechanically balanced by counteracting left-handed torques. This may be achieved by locally deviating from the default eight-base-pair crossover spacing design rules following previously discussed concepts[4], by reducing the average bases between crossovers to achieve the native 10.5 base pairs per turn. We thus designed and solved a cryo-EM structure of a refined variant of the Twisttower in which we eliminated the global twist deformations. As a result, we obtained a more orderly object with a more regular square-lattice structure (Fig. 2a, right).

### Global twist deformation in a honeycomb lattice.
The majority of other objects in our library were multilayer DNA origami in honeycomb-lattice packing. Nearly all cryo-EM maps that we determined for objects built using the default seven-base-pair strand crossover spacing prescribed by honeycomb-design rules displayed global right-handed twist deformations (Fig. 1a, f left, 1 g left, 1 h, Fig. 2b–f). The appearance of these twist deformations is noteworthy because the crossovers in seven-base-pair helical intervals create helical connections in a threefold symmetry that closely matches with the natural 10.5 base pairs per turn B-DNA twist density. Hence, in contrast to square-lattice designs, honeycomb designs do not impose a priori helical deformations and twist buildup is not necessarily expected[2,4]. The extent of the global twist deformation seen in our panel of honeycomb-lattice structures also decreased with the increasing cross-sectional area, consistent with the behavior we saw for a twist in the domains of the square-lattice Twisttower. The 126-helix bundle (Fig. 2b), which featured the widest cross section in our panel, did not display any detectable global twist. The slenderest object, the six-helix tube, exhibited the largest twist deformation, with a 90° total twist over 100 base pairs, in a design implementation that places strand breaks directly at crossover sites for improved folding yield (Supplementary Fig. S28)[21,22]. A second variant of the six-helix tube had a negligible global twist but instead curvature, and assembled with poor yield (Fig. 2c, bottom). This second variant was designed with legacy rules where strand breaks are placed systematically away from crossovers[2]. Both six-helix-tube variants use the same default honeycomb seven-base-pair crossover spacing. Our observations made with these two six-helix-tube variants highlight that not only folding behavior but also solution shape may sensitively depend on the internal design details, emphasizing the need for structural validation. We also designed and analyzed the structure of a third, further refined six-helix-tube variant featuring base-pair deletions every 21 bases. This third variant folds well and does not exhibit a global twist deformation nor curvature (Fig. 2c, top and Supplementary Fig. S28).

We previously developed the hinged-beam-like object (Figs. 1f, 2d, left) for positioning fluorophores and reactive groups[23], as well as for measuring forces between nucleosomes[24]. Here, we determined a cryo-EM map for this object and the map reveals a

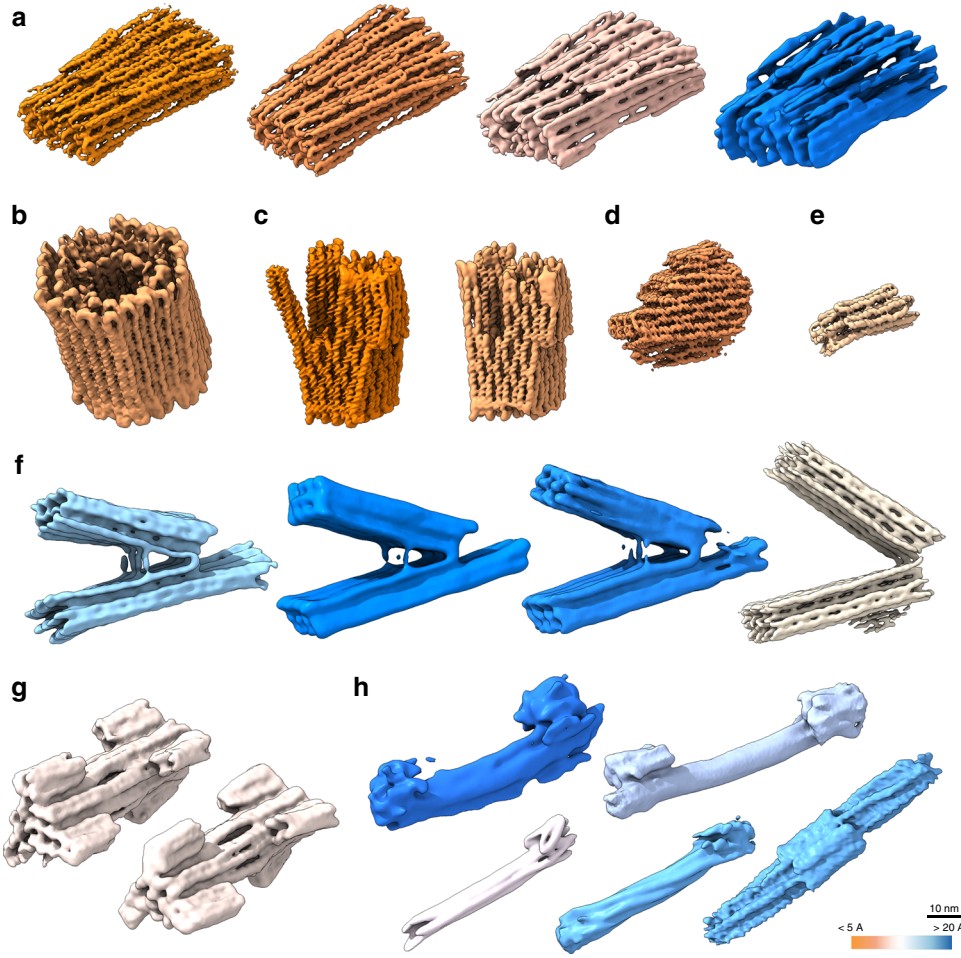

**Fig. 1 Twenty DNA origami cryo-EM solution structures drawn to scale.** All maps were determined in freestanding ice. **a** Four 48-helix Brick DNA origami objects. Left to right: 0, 1, 2, 4 unpaired thymidines added at each staple crossover, respectively. **b** A barrel-like 126-helix bundle DNA origami object one-pot assembled with two scaffold chains. **c** Two design variants of a multidomain square-lattice DNA origami called Twisttower. **d** A design variant (v2) of the previously reported Pointer[13]. **e** A 16-helix bundle DNA origami object assembled from a 1317 base-pair long-scaffold strand. **f** Four variants of a previously described hinged-beam-like object[23]. **g** Two variants of a dumbbell-like DNA origami object with a crossover-free handle segment. **h** Four six-helix bundle DNA origami objects, two with one asymmetric feature at one end (bottom), two with two asymmetric features at each end. Bottom right: a ten-helix bundle with one asymmetric feature in the middle. See Supplementary Figs. S1–20 for more cryo-EM imaging details. Strand diagrams for each object are supplied in Supplementary Figs. S43–S63.

pronounced right-handed twist deformation in the two beams of the object. Our cryo-EM map thereby invalidates the straight geometrical model that we previously assumed[23,24] to calculate point-to-point distances. Since the assumed versus actual point-to-point distance changes upon angle change between the beams are negligibly affected by the twist deformation, conclusions drawn in the previous work are not affected. We titrated the density of counter-twist-producing modifications necessary in the hinged-beam object to entirely remove the global twist deformations and solved cryo-EM structures for each design iteration (Fig. 2d, left to right), thus yielding a design variant that now does correspond to the previously assumed straight geometrical model.

**Empirical global twist correction**. Our set of cryo-EM solution structures furthermore allows constructing an empirical guide for estimating the expected global twist deformations, and for refining the objects to achieve negligible global twist deformations. To this end, we plotted the estimated polar moment of inertia (the torsional stiffness) versus the observed global twist per base along the helical direction in each cryo-EM map that we determined (Fig. 2e). The graph also gives the effective helical twist density that is imposed by design context (square or

honeycomb lattice) and the number of counter-torque-producing base-pair deletions or insertions that we installed (Fig. 2e). Researchers can estimate the polar moment of inertia based on the helical cross section of the planned object and then use to plot to read off the required average segment-length deviation from default honeycomb or square-lattice crossover spacing to approximately yield zero twist per base.

In order for the counter-torque-based design refinement to succeed, helices must be connected by crossovers in order to transmit the twist-countering torques. To illustrate this requirement, we built two variants of a dumbbell-shaped multilayer DNA origami featuring rotor-like indicator domains at either end (Fig. 2f). The central axis segment of the dumbbell consisted of 24 parallel, but unconnected helices. The cryo-EM structures that we solved for the two variants of the dumbbell overlapped closely and had the same twist deformation even though we installed differing local helical twist density in-between the rotors (Fig. 2f). This shows that due to the lack of crossovers in the dumbbell axis, torque transmission arising by helical under- or overwinding is not effective.

Since we just saw that crossovers are necessary to transmit the torques, we wondered whether strains originating from crossovers

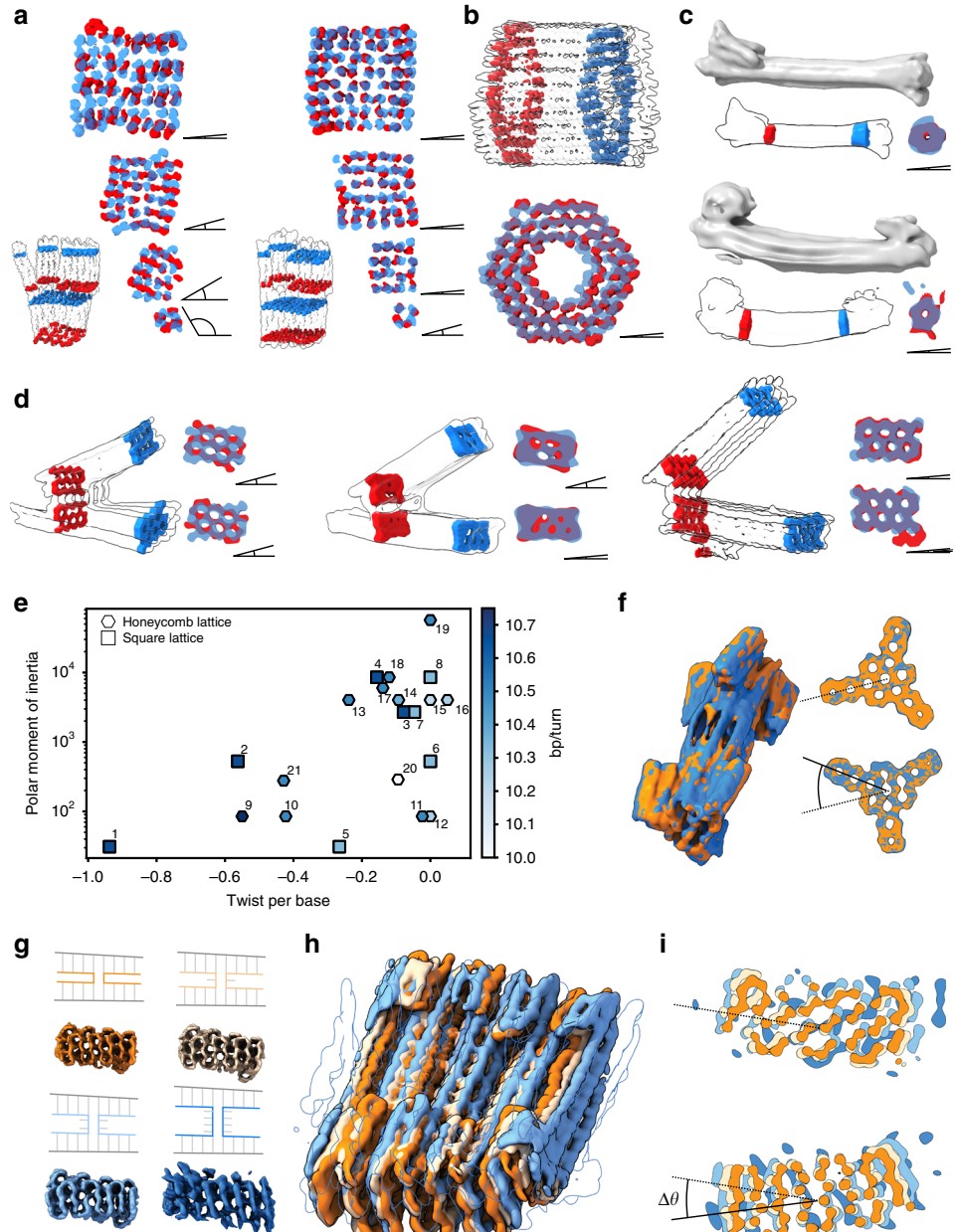

**Fig. 2 Global shapes: twist, torsional stiffness, twist removal, and crossover strain. a** Left versus right: cross-sectional slices of the cryo-EM maps of the four domains of the default Twisttower and twist-corrected Twisttower design variants, respectively. The angle symbol visualizes the amount of twist between the blue and red layers for a given subpart. Inset: the outline gives full cryo-EM map, the position of the cross-sectional slices are as indicated. **b**, **c** As in (**a**) but for the 126-helix bundle and for two six-helix-tube design variants. **d** Outlines and cross-sectional slices in cryo-EM maps of three twist-titrated variants of the hinged-beam object. **e** Symbols give a polar moment of inertia over the observed global twist per base pair in the 20 DNA origami cryo-EM maps. Colors indicate the effective helical twist density as imposed by design. 1–4: Twisttower 2 × 2, 4 × 4, 6 × 6, and 8 × 8 domains; 5–8: twist-corrected Twisttower 2 × 2, 4 × 4, 6 × 6, and 8 × 8 domains, respectively; 9–12: six-helix tubes v2, v3, v4, and v1, respectively; 13–16: hinged-beam-like objects v1, v2, v4, and v3, respectively; 17: 42-helix brick-like; 18: 48-helix brick; 19: 126-helix barrel-like; 20: 16-helix bundle; 21: ten-helix bundle. Source data are provided as a Source Data file. **f** Overlay of the cryo-EM maps determined from two dumbbell-like DNA origami objects. **g** Four DNA origami 48-helix-brick design variants with 0, 1, 2, 4 unpaired thymidines at all staple crossovers. **h** Overlay of the cryo-EM maps of the 48-helix-brick variants. **i** Cross sections of all four variants. Coloring as in (**g**).

constitute the root cause of twist in honeycomb objects. If this were true, making crossovers floppier should reduce twist deformations. To test this hypothesis, we designed and analyzed a set of multilayer DNA origami 48-helix-brick variants in which we added unpaired thymidines at all staple-strand crossovers (0T, 1T, 2T, and 4T). All variants displayed right-handed global twist deformations of comparable extent (Fig. 2g–i), which suggests that strains produced at crossovers do not cause the twist. Since helical

details could be discerned in the majority of the helices of the map obtained with the 0T variant, we could determine the handedness of the global twist deformation based on the fact that B-DNA is right-handed. Helical details were lost in the EM maps already upon adding one T per crossover, but we presume that the twist deformations observed for the variants with more T's at the crossovers have the same handedness as the 0T variant. Whereas the global twist remained unaffected, adding the thymidines made

the objects swell (Fig. 2h, i). The interhelical spacing and the effective diameter of helices increased with T addition, which we attribute to the increasing fluctuations enabled by the increasingly floppy interhelical junctions.

Previous work indicated that the spatial distribution of single-strand backbone discontinuities (nicks), which could constitute torsionally weak points in double-helical domains, influenced the twist angle of six-helix tubes as seen by AFM adhered to a solid support[25]. To test the relevance of this design parameter in solution, we determined cryo-EM maps of variants of 42-helix brick-like objects in honeycomb-lattice packing (Supplementary Figs. S21–26). In one variant, the staple-strand nicks were distributed randomly; in the second variant, the nicks were aligned on a set of (virtual) cross-sectional planes. The third variant was like the second variant but had additional unpaired thymidines at the nick sites for UV-point welding[26]. The resulting cryo-EM maps all overlapped closely and featured identical global twist deformation. Hence, nick-site distribution appears inconsequential with respect to twist buildup. Previous computational studies[17,27,28] indicate that for lattice-based DNA origami, the stacking interaction at nick sites might be strong enough to compensate for the missing backbone connection. Introducing gaps might have a greater influence on the overall twist compared to nicks. We note that the irradiation of the objects with UV light eradicated the twist deformation, based on cryo-EM maps that we solved after exposing the samples to 310-nm light (Supplementary Fig. S29), which corroborates previous findings with single-layer structures seen adhered on solid supports[29].

**Revealing higher-resolution features**. We used 3D classification with a large number of classes to uncover the spectrum of shapes sampled by a given DNA origami object. Exemplarily, we found that the Twisttower samples an ensemble of conformations in which the different domains move relative to each other by bending and twist deformations at the domain interfaces. In particular, the $2 \times 2$ and the $4 \times 4$ domains show relative displacements with up to 12-nm (extreme to the extreme) amplitude (Fig. 3a and Supplementary Movie M1). Furthermore, we also observed breathing motions in which the entire helical lattice expands and shrinks, as exemplified in data obtained with the Pointer-v2 object (Fig. 3b and Supplementary Movies M2, M3)[13]. The Pointer v2 also exhibited other types of structural fluctuations akin to domain motions. These fluctuations are design-specific and depend on the global shape as well as the topology of the nanostructures. As these motions are driven by the thermal fluctuation of the individual helices, they are also dependent on temperature and salt- and buffer conditions, as well as the overall folding quality of the nanostructure ensemble. Folding defects, either caused by partially unhybridized or defect oligonucleotides, can locally influence the mechanical properties of a helical segment and distort the global shape or act as a hinge for a domain motion. Hence, DNA origami are not rigid objects, instead, they display substantial structural heterogeneity. The heterogeneity may be coarsely classified as relative domain motions similar to those seen in proteins and as helical lattice breathing, which does not exist in proteins. Ignoring these motions and reconstructing the structure of a target object as a single static entity will blur high-resolution detail that might have been present in the data.

**Multibody refinement in the presence of internal motions**. To systematically deal with internal motions, we adapted focused refinement methodologies[30]. We demonstrate the efficacy of this approach exemplarily with the results obtained with the Twisttower (Fig. 3c). Figure 3c, top left, shows the result of a 3D

refinement that assumes the whole object as one rigid body. While the quality of this cryo-EM map is already superior to any other DNA origami structure published thus far, the helical interfaces at the periphery of the object and substantial portions of the $4 \times 4$ domain remain blurred. The $2 \times 2$ domain can hardly be discerned at all. To deal with relative domain motions, we divided the Twisttower object into its domains (i.e., the $2 \times 2$, $4 \times 4$, $6 \times 6$, and $8 \times 8$ regions) and used multibody refinement[30] to separately reconstruct the 3D structure of these domains pretending that now the domains, but not the entire Twisttower, were rigid bodies. The resulting 3D maps for the separate domains offered improved detail. In particular, helical grooves can now be seen in all regions of the domains and in particular at the periphery (Fig. 3c, insets on the right). A Frankenstein map of the Twisttower (Fig. 3c, bottom), which is generated by merging the maps from the different focused refinements, allows appreciating that now virtually all parts of the Twisttower are resolved with high detail (see also Supplementary Fig. S30). We applied this analysis also to the twist-corrected variant of the Twisttower (Supplementary Fig. S31) and the hinged-beam-like object v4 (Supplementary Fig. S32). To show the efficacy of the multibody refinement even when the object does not have clear domains, we applied the multibody approach to three brick-like objects (Supplementary Figs. 3d, S33–S35) and the 126-helix bundle (Supplementary Fig. S36). Simply dividing the whole object into smaller rigid bodies (Fig. 3d, right) enabled reconstructing all regions of the objects in greater detail including the peripheries (Fig. 3d, top vs. bottom).

**Focal scanning refinement**. The maps can still be further improved by using a focal scanning two-body refinement approach that allows dealing with helical lattice-breathing motions. To this end, we further divided the previously defined domains into two segments, one being a focal cross-sectional element (e.g., a domain consisting of $2 \times 2$ double helices, each 32-base-pairs long) to be reconstructed with higher detail (Fig. 4a), and the other comprising the rest of the domain, surrounding the focal element. We thus obtained improved level of detail in the focal element (Fig. 4a, left vs. right). This two-rigid-body refinement approach can then be iterated by scanning the small focus area across the entire object (Fig. 4b). The results from these separate localized refinements may be inspected separately, each revealing a portion of the target structure with high detail, and all of them may again be combined into a Frankenstein map (Fig. 4b, bottom).

Using the focal scanning refinement procedure, we were able to reconstruct details of $2 \times 2$ parts of the Twisttower object with a global resolution of 4.3 Å and local resolutions below 4 Å (Supplementary Fig. S37 and Supplementary Movie M4). At this level of detail, not only helical grooves can be seen but also the molecular details of the helices emerge. Backbone phosphates manifest as bumps (Fig. 4a, right); the covalent single-strand versus double-strand backbone connections between helices can be recognized as thin bonds and discriminated from each other (Fig. 4c, d, respectively). Nicks that lack phosphates manifest as depressions or kinks in the groove boundaries (Fig. 4e). The thus-refined maps now also deliver detail at the very periphery of the objects, including density from single-strand tails and peripheral crossovers (Fig. 4f). Obtaining this high level of detail is a major step forward that now allows designers to inspect and refine their designs with base-pair-level interventions.

**Pseudo-atomic model construction**. The availability of high-quality structural data immediately presents a challenge, which is the interpretation of the map with atomic models and forming a

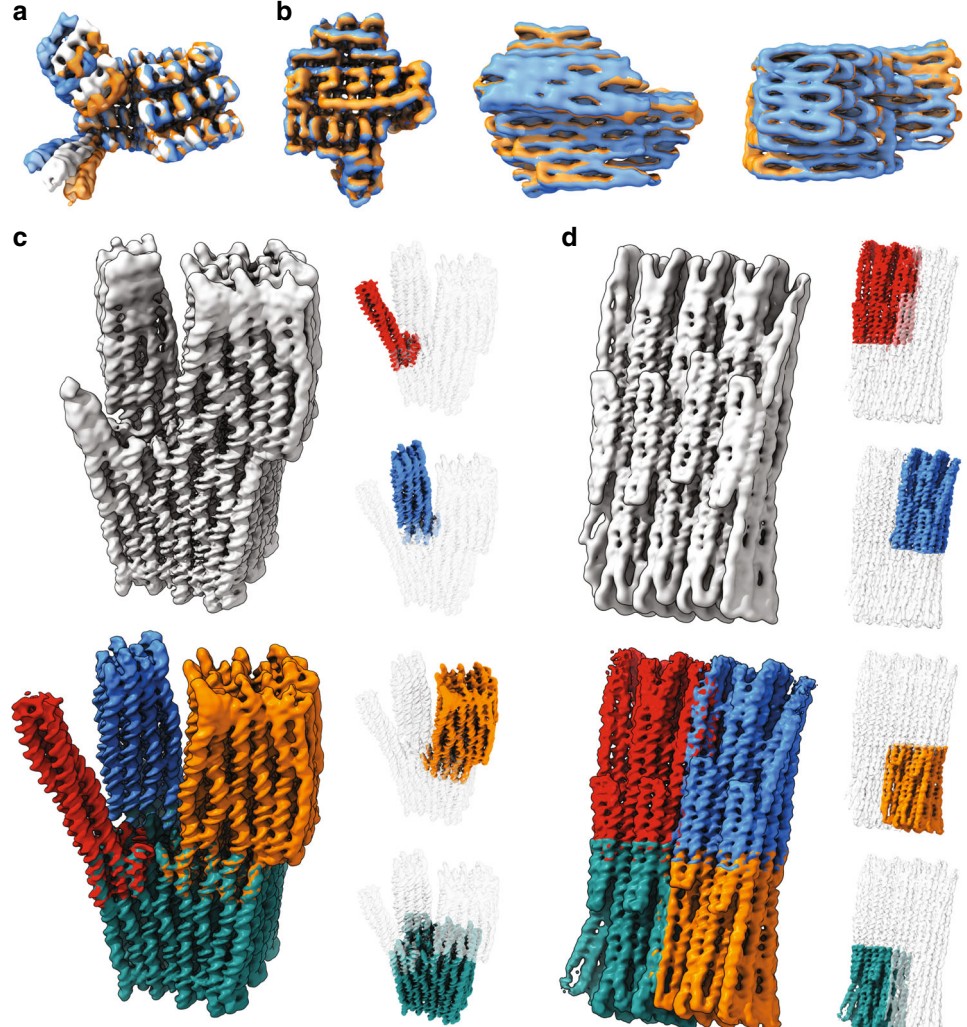

**Fig. 3 Domain motions and lattice breathing in DNA origami. a** Overlay of three exemplary 3D classes determined from the Twisttower dataset. **b** Overlay of exemplary 3D frames of a principal component analysis (PCA) of the heterogeneity in the Pointer-v2 dataset. **c** Top left: cryo-EM map determined treating the Twisttower as one rigid body. Right: cryo-EM maps of subbodies analyzed using multibody refinement. Bottom left: Frankenstein map. **d** As in **c**, but for a 48-helix-brick object, with an arbitrary division into four subbodies.

link to the actual design. To build such a model, a suitable initial model is required, which can then be systematically corrected to optimally fit into the electron density. For creating initial models, we used the atomic-detail predictions produced by ENRG-MD[31]. Fitting the initial model into the EM densities is complicated by the pseudo-periodic lattice structure of the DNA origami EM maps. Many local minima exist in which a fit can get stuck. To carry out the fitting, we used molecular dynamics flexible fitting[32], in which the electron-density map acts as an attractive force field, and developed a cascaded relaxation[33] protocol, in which we relieved restraints sequentially from the ENRG-MD force field as the correlation between measured EM map and the fitted atomic model improved (see "Methods"). Figure 5a shows snapshots of our cascaded relaxation procedure (see also Supplementary Movie M5) for a region within the Twisttower object. Our methodology allows pseudo-atomic model construction in a semiautomated fashion within ~12 h compute time on a standard desktop computer, which may be compared to the several weeks it took to manually construct the atomic model of the previously reported Pointer object[13]. We constructed pseudoatomic models for six different objects (Fig. 5b). The fitting was validated using Fourier shell correlation against the cryo-EM half-maps

(Supplementary Fig. S38) and the models show good cross-correlation with the cryo-EM maps (Supplementary Table S3). We also refitted the previously reported Pointer cryo-EM map with our semiautomated approach, and the computed atomic model closely matches the manually constructed one[13] (Supplementary Fig. S39).

**Context-based zoning and cropping of cryo-EM maps**. We developed a viewer tool[34] to form a link between the experimentally determined cryo-EM map, the fitted atomic model, and the strand diagram prepared by the designer to build the DNA origami object under study. The tool links the actual geometry of the object as seen in the cryo-EM map and annotated by the fitted atomic model in terms of cartesian atomic coordinates with the connectivity index system used by designers in DNA origami-strand diagrams (helical and base indices). The tool allows cropping or zoning the map to dissect it into elements of interest and displaying it together with the corresponding segments of the atomic model. For instance, we used the tool to systematically segment all measured maps into the constituent lattice layers to visually inspect the maps and the quality of the atomic model that was fitted to the map (Supplementary Fig. S40). We used these

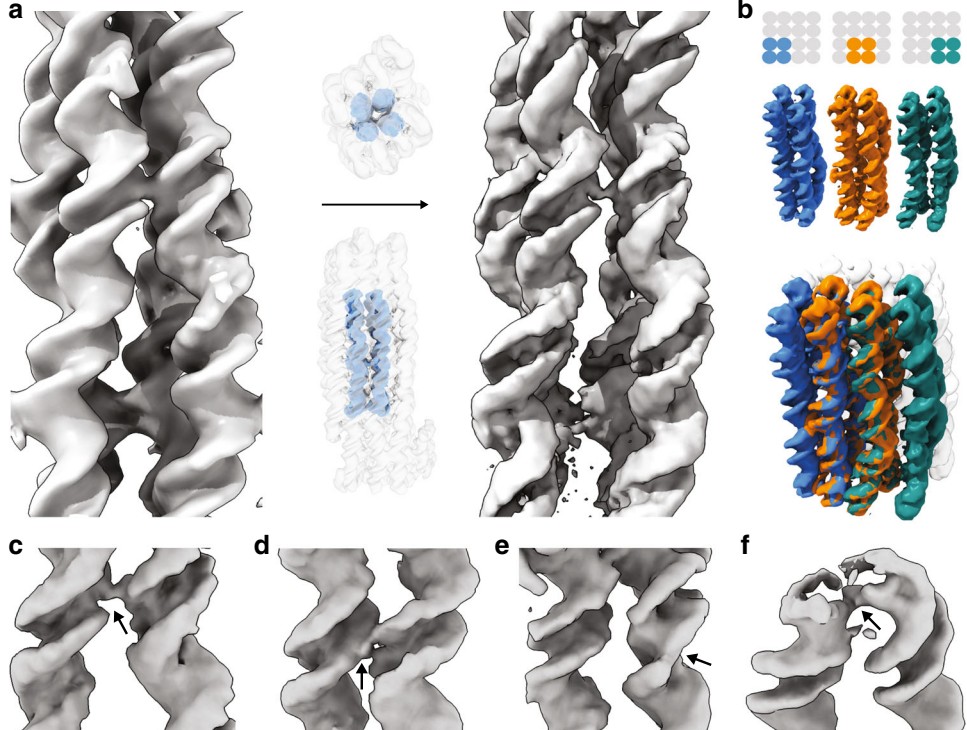

**Fig. 4 Scanning-focused refinement. a** Left: exemplary region of interest in the 4 × 4 domain in the Twisttower cryo-EM map. Right: same region, after two-body refinement using the two bodies indicated in the middle. **b** Schematic illustration of the iterative application of the procedure in (**a**) over the entire object. Bottom: Frankenstein map, created by combining the focal elements. **c–f** Left to right: arrows give examples of high-resolution features revealed after scanning-focused refinement: single-strand phosphate backbone bond, double-strand phosphate backbone bond, depression at a strand-break site, peripheral single-strand crossovers, and density from poly-T tails at a helical interface.

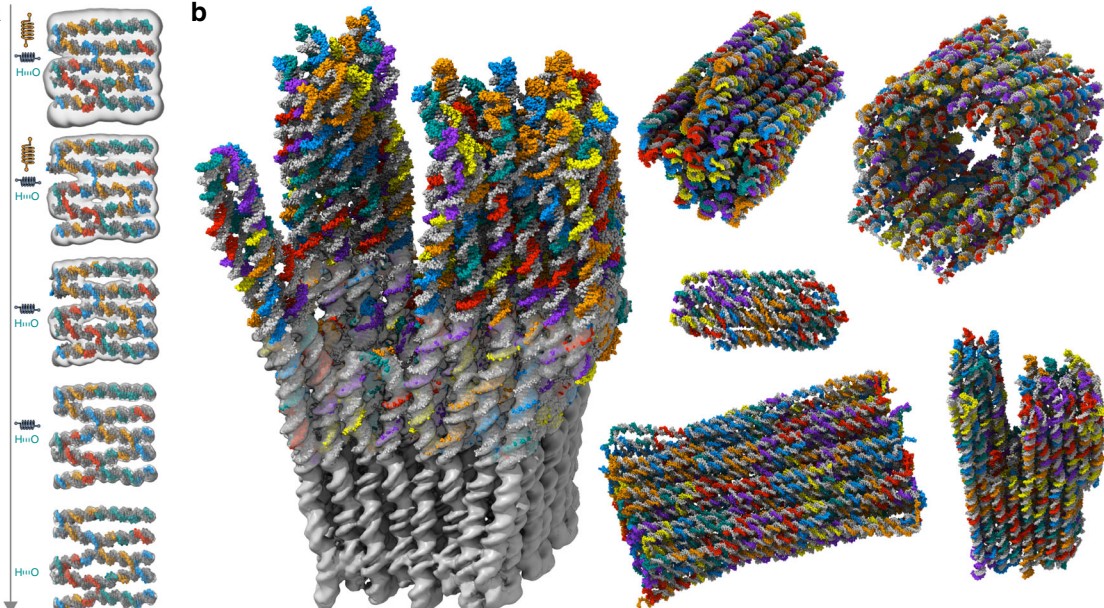

**Fig. 5 Semiautomatic atomic model construction. a** Schematic illustration of the cascaded relaxation procedure. Markers along the arrow represent secondary restraints applied to the structure: interhelical (orange, vertical springs), intrahelical (blue, horizontal springs), and hydrogen bonds of the Watson–Crick pairs (H–O). **b** Atomic models derived from fitting to cryo-EM maps: Twisttower (overlay with map), Pointer-v2, 126-helix bundle, 16-helix bundle, 48-helix brick, and twist-corrected Twisttower. Scaffold atoms are depicted in gray, staple strands in color.

dissections to reveal unexpected or misfolded structural features that otherwise would have been hidden in the depths of the full cryo-EM maps (Supplementary Fig. S41).

## Discussion

In conclusion, our set of cryo-EM structures covers a range of different DNA origami designs and provides insight into structural changes that follow from subtle variations of these designs. Thereby, our dataset (see Supplementary Table S4 for EMBD and PDB IDs) provides the constraints needed for parameterizing computational structure prediction methods, whether coarse-grained or with atomistic detail[17,31,35,36]. Exemplarily, we computed the deviations of atomistic ENRG-MD predictions[31] from our atomic models, which were all >9.6 Å (RMSD) (see Supplementary Table S3) and thus clearly above the resolution of the experimental maps used for fitting. The strongest deviations between experimental data and prediction occurred at locations where the design deviates from idealized lattice rules, e.g., at sites with omitted crossovers or with sudden changes in the helical cross section (Supplementary Fig. S42).

Our set of atomic coordinates obtained from fitting six DNA origami electron-density maps comprises ~100,000 base-pair coordinates, roughly equally distributed over all the possible base-pair step sequences, which may be compared to ~65,000 DNA-only base-pair coordinates currently available in the protein data bank (PDB) (Feb 2020). The depth of data could also allow mining for sequence–structural relationships in B-DNA that could help advance DNA nanotechnology from the current sequence-agnostic design methods to more refined approaches that optimize sequences for target backbone coordinates, akin to strategies used in de novo protein design[37].

Our results underline the importance of structure validation in solution. For example, the presumed geometry of one of our own previously reported objects turned out to be not quite correct. It is likely that the actual geometries of many other previously reported objects deviate from the idealized expectations. This is particularly relevant in applications where twist deformations could affect the results (e.g., refs. [38–41]) and with objects having slender cross sections such as six-helix tubes. In fact, six-helix tubes in honeycomb-lattice packing are popular objects that are used in a variety of contexts, ranging from NMR-based structural analysis[42] and liquid crystals[41] to single-molecule manipulation[43,44]. Our solution structures of six-helix tubes revealed a strong dependence of their shapes on design details.

We also demonstrated how iterative cycles of design and validation can help to correct an object to meet desired specifications, highlighting the programmability of DNA nanotechnology. Beyond validation of global shapes, we are convinced that revealing high-resolution features of DNA origami structures as we showed here with our focal scanning refinement (Fig. 4) will open new possibilities for the field. Now, researchers can zoom into regions of interest of a DNA origami chassis to iteratively refine them, for example, to tune the relative position and orientation of functional moieties or to model reactive centers.

## Methods

**Sample preparation**. The reaction mixtures contained homemade scaffold DNA, purchased staple oligonucleotides (Eurofins MWG and IDT), and folding buffer (1 × FOBx) at pH 8, including 5 mM Tris, 1 mM EDTA, 5 mM NaCl, and × mM MgCl$_2$ (see Data D1, Supplementary Table S5). The mixtures were subjected to 15 min of constant heating at 65 °C followed by a stepwise thermal annealing ramp using a Tetrad thermal cycling device (MJ Research, now Bio-Rad). The folding products were purified and concentrated using PEG purification and filter purification/concentration. The used type and concentration of scaffolds and oligonucleotides, concentration of MgCl$_2$, annealing ramps, and purification/concentration protocols depended on the type of structure (see Supplementary Table S5). The PEG purification[22] was performed by mixing the folding reaction in

a one-to-one ratio with a 15% PEG 8000, 5 mM TRIS, 1 mM EDTA, and 500 mM NaCl solution, and centrifuged for 30 min at 20,000 rcf. Afterward, the supernatant was discarded and the pellet dissolved in 1 × FOB. For the filter purification/concentration, the sample was diluted with 1 × FOB to a final MgCl$_2$ concentration of 5 mM. The Amicon Ultra 0.5-ml, 50-kDa cut-off filters (Millipore) were rinsed with 1 × FOB5. The sample was added to each filter and subjected to a centrifugation step at 10 k rcf for 5 min. After several washing steps consisting of removing of the flow-through, refilling of the filters to 500 μl with 1 × FOB5, and a centrifugation step, the filters were placed upside-down in fresh tubes and subjected to another centrifugation step.

**HPLC purification**. Excess staple DNA strands were removed from the reaction mixture by performing one round of polyethylene glycol (PEG) precipitation. The resulting pellets were dissolved in HPLC buffer (1 mM EDTA, 5 mM TrisBase, and 200 mM NaCl, pH 8) containing 5 mM MgCl$_2$. Then, we subjected the sample to HPLC (Agilent Technologies 1260/1290 infinity) using the column (Agilent Bio SEC-5: 5 μm, 2000 A, 21.2 × 300 mm) at a flow rate of 2 ml/min and collected fractions of the monomer peak (30–35 min). Due to dilution of the sample, we used ultrafiltration (30 K Amicon Ultra-15 mL from Merck Millipore) to concentrate the sample and to exchange the buffer to folding buffer (1 mM EDTA, 5 mM TrisBase, 5 mM NaCl, and 5 mM MgCl$_2$, pH 8).

**UV-point welding**. For UV-point welding[26], we used a 300-W xenon light source (MAX-303 from Asahi Spectra) with a high-transmission band-pass filter centered around 310 nm (XAQA310 from Asahi Spectra). We used a light guide (Asahi Spectra) to couple the light into the sample by placing it directly on top of a 0.65-ml reaction tube. Unless otherwise indicated, the brick-like samples were irradiated for 120 min. Samples were irradiated in folding buffer (5 mM Tris, 1 mM EDTA, and 5 mM NaCl), including 30 mM MgCl$_2$, unless otherwise stated. After irradiation, the buffer was exchanged to folding buffer, including 5 mM MgCl$_2$.

**Cryo-grid preparation and image acquisition**. The purified and the concentrated sample was applied to glow-discharged C-Flat 2/1 4 C (Protochips) or C-Flat 1.2/1.3 4 C grids (Protochips) and plunge-frozen using a Vitrobot Mark IV (FEI, now Thermo Scientific) at the following settings: temperature of 22 °C, the humidity of 100%, 0-s wait time, 2–4-s blot time, −1 blot force, and 0-s drain time (Supplementary Table S2). For the Pointer object and the 16-helix bundle, homemade graphene oxide-coated holey carbon grids were used. Graphene oxide dispersion in H$_2$O (Sigma) was diluted to 0.2 mg/ml in H$_2$O and spun at 300 g for 30 s to remove large aggregates. Quantifoil R1.2/1.3 holey grids were glow-discharged for 1 min, and 3 μl of the graphene suspension was added to the grids for 1 min. Grids were subsequently blotted briefly using Whatman No1 filter paper and washed three times in 20-μl drops of H$_2$O (twice on the graphene side and once on the reverse side). Grids were then used for plunge-freezing without further treatment[45]. The data were acquired on a Titan Krios G2 electron microscope operated at 300 kV equipped with a Falcon 2, later upgraded to a Falcon 3, direct detector using the EPU software (FEI, now Thermo Scientific). The acquisition parameters for the individual data sets are summarized in Supplementary Table S2.

**Cryo-EM data processing**. The image processing was performed in Relion 2[46] and 3[30]. The micrographs were motion-corrected and contrast-transfer function estimated using MotionCor2[47] and CTFFIND3 and CTFFIND4[48], respectively. The particles were picked using the Relion and Cryolo[49] autopickers. For the autopicking procedure in Relion, a few thousand particles were manually picked and subjected to reference-free 2D classification to create templates. The autopicked particles were extracted from the micrographs and subjected to multiple rounds of 2D and 3D classification to remove falsely picked grid contaminations and damaged particles and to address structural heterogeneity. A refined 3D map was reconstructed using a low-resolution initial model created in Relion. The particles were polished (per-particle motion correction and dose weighting), and a polished 3D-refined map was reconstructed. The map was post processed using a low-pass-filtered mask to calculate the FCSs and estimate the global resolution. Based on the local resolution estimation implemented in Relion, the map was also locally low-pass filtered.

**Multibody analysis**. The procedure was performed using multibody refinement in Relion 3[30]. The consensus maps were divided into subset parts using the eraser tool in UCSF Chimera[50]. The parts were low-pass filtered, binarized, and multiple layers of soft-edge voxels were added to create the masks for multibody refinement. The multibody-refined maps were post processed using low-pass-filtered masks to calculate the FCSs and locally low-pass filtered based on their estimated local resolution. For each component of the principal component analysis, particles were distributed into ten equally populated bins according to their eigenvalues to create maps representing the motion of the respective components. For the subsequent focused two-body refinement of the Twisttower, the particles were re-extracted at the centers of projections of the four domains, with smaller subarea boxes but original pixel size from the micrographs, subtracting the signal from the other bodies. The resulting smaller boxes allow more efficient processing. From the resulting four subtracted particle sets, refined maps of the individual bodies were

reconstructed. To focus on a small subvolume, the map was divided into a map containing the region of interest and a second map containing the rest using UFSF Chimera. To address the remaining motion, multibody refinement with low-pass-filtered, soft-edged masks was performed instead of the simpler approach of partial signal subtraction and focused refinement.

**Model construction**. The initial pseudoatomic models were calculated using the idealized coordinates provided by ENRG-MD[31], with the caDNAno 2.3 strand diagram[6] and the nucleotide sequence as input. Models were manually pre-aligned with the cryo-EM maps using VMD version 1.9.3_MacOSX[51]. Approximate alignment of the center of mass and the helical direction is sufficient, as the procedure is capable of realigning the model. Before starting the fitting procedure, the steepest-descent energy minimization of up to 4800 steps was used to improve the model's geometry. All simulations were performed using the program NAMD 2.12_Linux[52] and the CHARMM36[53,54] force field for nucleic acids. The VMD packages mdff_0.5 and volutil_1.3 were used to prepare the grid-based potentials from the cryo-EM maps. All simulations were performed in the NVT ensemble, used periodic boundary conditions, smooth truncation of Lennard–Jones and short-ranged Coulomb interactions at 10 Å, and a switch distance of 8 Å. No PME long-range corrections were applied and Langevin forces for all nonhydrogen atoms with damping of 0.1 1/ps were used to keep the temperature constant. Simulations were performed in a vacuum with dielectric constant 1. For the Twisttower, the twist-corrected Twisttower, and the barrel-like 126-helix bundle, composite maps from the multibody refinement were used for fitting.

**ENRG-MD-driven cascaded flexible fitting**. The cascaded relaxation procedure is based on molecular-dynamics flexible fitting (MDff)[32,55] and consists of three phases. First, the helices of the initial model are globally aligned. MDff is performed with a weight on the EM density of 0.3 kcal/mol. To avoid the strong local minima generated by the lattice, the model is fitted to a series of cryo-maps of sequentially improving resolution, an approach called cascading flexible fitting (cMDff)[33]. Maps are low-pass filtered using a gaussian blur of up to 22 Å. This initial cascade consists of eight maps with an overall resolution of 22–10 Å. Each map is used to perform 12,000 steps of MDff. As the strong deformations of the model during these early stages can significantly distort the geometry of the model, the elastic network (EN) provided by ENRG-MD[31] is used throughout this process. The harmonic bonds not only provide extra stability, but they also significantly speed up the dynamics of the structure. Consequently, the initial cascade can correct strong misalignment of the map and initial model, although large deviations might necessitate more time to align properly. This is the case for the Twisttower, where the different domains cannot be aligned with the initial model correctly. Additionally, the 2 × 2 sections are twisted by more than 90 degrees, necessitating a prolongation of the early alignment process. For the Twisttower, the cascade started at 24 Å and 60,000 steps.

During this general helical alignment step, local deformations, especially close to holiday junctions, can occur. These deformations are a consequence of the nonglobal nature of the MDff minimization combined with the restraints of ENRG-MD, which do not permit large structural deviations from B-DNA. These harmonic restraints can be categorized into long-range interhelical, short-range intrahelical, and bonds connecting the Watson–Crick base pairs. To resolve these local deformations, the grid-based potential is switched off for 12,000 steps, locally relaxing the structure using all EN restraints[31]. Then, a second cascade is performed using eight maps of overall resolution from 16 Å to the final resolution of the map. MDff is performed with 6000 steps for each map. As the model is already closer to the hypothetical solution structure, the EN is no longer required to suppress strong deformations. Reducing the number of harmonic bonds increases the sampled face space of the procedure and improves the model's accuracy. Consequently, long-range bonds of the EN are turned off for maps with a resolution of 14 Å or better. Finally, the model is fitted against the original map for 12,000 steps with only base-pair-restraining bonds still in place. Afterward, the weight on the EM density is increased to 1.0 kcal/mol for 18,000 energy-minimization steps. The selection of the different harmonic bonds is performed in the context of the strand diagram.

**Model validation and stereochemical quality**. To assess the quality of the fit, the masked cross-correlation coefficient (ccc) was calculated using the mdff_0.5 package in VMD[51] (Supplementary Table S3), using the map resolution reported in Supplementary Table S3. The masks were generated with the viewer tool[34]. Additionally, helical properties like helical rise, twist, and base-pair orientation were calculated and monitored throughout the fitting procedure to assess outliers and big changes in the model's geometry. We also used a half-map-based validation approach to prevent overfitting[56]. Fourier Shell Correlation for the final fitted model and the experimental cryo-EM map was calculated using Relion[46] (see Supplementary Fig. S38). The simulated maps were generated in VMD.

**Reporting summary**. Further information on research design is available in the Nature Research Reporting Summary linked to this article.

## Data availability

All maps and fitted models that support the findings of this study are available in the EMDB[57] and in the Protein Data Bank (PDB)[58], respectively: EMD-11379, EMD-11378, EMD-11881, EMD-11170, EMD-11367, EMD-11387, EMD-11343, EMD-11344, EMD-11345, EMD-11355, EMD-11351, EMD-11352, EMD-11353, EMD-11354, EMD-11346, EMD-11348, EMD-11349, EMD-11350, EMD-11159, EMD-11168, EMD-11294, EMD-10993, EMD-11295, EMD-11296, EMD-11297, EMD-11298, 7ARV, 7ARY, 7ARE, 7AS5, 7ARQ, 7ART. Identifiers with corresponding names of the structures are also listed in Supplementary Table 5. Raw cryo-EM data are available from the corresponding author upon reasonable request. Source data are provided with this paper.

## Code availability

The viewer tool for context-based zoning and cropping of cryo-EM maps is free open-source software under the GNU Public License version 3 and can be downloaded from https://github.com/elija-feigl/FitViewer [34]. It is developed as a Python-based Jupyter Notebook[59]. Atomic model trajectories and coordinate files are read/written by the python package MDAnalysis 0.20.1[60], cryo-EM data are handled by the package mrcfile 1.1.2[61]. A custom version of the package autodesk/nanodesign (https://github.com/elija-feigl/nanodesign_dietz) is used for reading the caDNAno strand diagrams.

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

## Acknowledgements

This work was supported by a European Research Council Consolidator Grant to H.D. (GA no. 724261), the Deutsche Forschungsgemeinschaft through grants provided within the Gottfried-Wilhelm-Leibniz Program and the SFB863 TPA9 Project ID 111166240 (to H.D.), and the UK Medical Research Council (MC_UP_A025_1013 to S.H.W.S.). Additional support came from the Max Planck School Matter to Life (a joint program of BMBF and Max Planck Society) to H.D. and E.F.

## Author contributions

H.D. designed the research, M.K., F.K., and E.F. performed research. M.K. and F.K. performed cryo-EM measurements and data processing. E.F. constructed atomic models. B.N. supported sample preparation and data analysis. E.W. provided the dumbbell samples. J.F provided hinged-beam designs. T.G. provided brick variants. P.S. provided the 10-helix tube and 16-helix bundle samples. M.H. provided the scaffold. T.M. made and analyzed the Pointer v2. S.H.W.S. supervised Pointer-v2 research and contributed analysis tools. All authors commented on the paper.

## Funding

## Competing interests

The authors declare no competing interests.
