## [Peer Review File · Nature Communications]

Reviewers' Comments:

Reviewer #1:

Remarks to the Author:

The authors present a very comprehensive and impactful study that elucidates several important structural features of brick-like DNA origami objects using 3D cryoEM, with rigorous, systematic experimentation, and innovative structural analysis and modeling that should prove very valuable to the nanotechnology field.

Specifically, they first examine how global twist in square lattice objects changes with cross-sectional size, and how it can be removed by altering crossover positions. Interestingly, they also observe global twist deformations for 'nearly all' honeycomb objects, which is unexpected given the canonical crossover spacings that match the natural twist of B-form DNA. They also apply their twist correction approach to a previously published FRET ruler. They test the hypothesis that tight/rigid crossovers induce this global twist, by introducing additional bases that relax the crossovers without effect, which demonstrates that this is not the origin. They then test nick placement too and demonstrate that this (random versus planar positioning) does not explain the origin of the global twist, or impact its relaxation. They then apply UV light exposure and find that this does eliminate the global twist. They go on to apply focused refinement to obtain impressive increased resolution in "rigid" subdomains, applied to a tower. Using a focal scanning approach they then are able to increase resolution to 3.8 Angstroms, which for the first time in the origami field offers true base-pair-level structural details to be resolved, which is of clear importance to application to structural as well as functional studies. Finally, they use an atomic modeling based approach to generate best fitting models efficiently, which should prove additionally very helpful for the interpretation of such datasets, and also provide a viewer for interactive viewing of structural data.

Given the numerous important and impactful contributions of this study, I recommend publication after addressing the several minor, yet also important, points below:

> Given that the Rothemund rectangle is one of, if not the, most widely used DNA origami objects in the DNA nanotechnology field for a wide range of applications, both on surfaces and in solution, and it is commonly assumed to be flat, the finding that the Rothemund rectangle may in fact exist in highly warped and heterogeneous sub-states is of major significance; This result would ideally be more prominently displayed in the main figures, or at least mentioned as an important finding, expanding on its important consequences on applications of this object in nanotechnology, rather than only mentioned in passing and shown in Supplement

> In the first, square lattice set of designs, it is noted that global twist is observed, but it is not specified whether it is right- or left-handed; on L82, it is then stated that global twist was eliminated, but it is not stated how; was this using modeling such as CanDo, or expert intuition, or another approach? While twist modification has previously been performed by the Dietz lab, and others, on numerous occasions, it would still be helpful to specify the approach to the reader here

> Similarly, for the second set of honeycomb designs, the results on global twist are very interesting, but again the directionality of the twist appears to be missing, and it is unclear on L109 how the 'refined variant' that is designed to be straight was designed/manipulated, which would be helpful for the reader and future designers to understand

> While nick placement is one possibility for eliminating twist, this assumes that nicks "release" torsional stress that's propagated along the duplex, which is not necessarily the case; Frank-Kamenetskii has shown that DNA basepairs still stack with significant free energy when nicks are present, and that "gaps" (i.e., deletions of bases) are required to eliminate this stacking; Kim et al., NAR 40: 2862 (2012) and Pan et al., 45: 6284 NAR (2017) examined this effect computationally in CanDo; While the present authors do not appear to have tested this possibility

directly by introducing “gaps” (which were also used in Wang et al., JACS 138: 7733 (2016)), their UV results support this “gap” hypothesis, since base deletion including strand rupture is likely to occur, each of which would eliminate the propagation of torsional stress along the duplex; While certainly not essential to test this experimentally in the current work, it would appear appropriate to include discussion of gaps along the lines of the above works as an additional useful hypothesis to test

Reviewer #2:

Remarks to the Author:

I think this is a great paper that reveals the structures of DNA origami complexes with nucleotide resolution, using solution cryo-EM and simulation. The reconstructed models with up to 3.8 Å local resolution clearly revealed details such as helical grooves, single-stranded versus double-stranded crossovers, backbone phosphate positions, and single-stranded breaks, along with fast semi-automated pseudo-atomic model building based on MD, this combined method will greatly help the inspections and refinements of various user-defined DNA nanostructures. Importantly, this is very much a step in building structural data bank of DNA nanotechnology, similar to the protein data bank. The work is beautiful! I recommend the acceptance without delay.

1. It is better to add a comparison table in SI with previous resolution data to clearly show that current quality of cryo-EM maps is superior to any other DNA origami structure published thus far.

2. Line 52, the authors found that “Rothemund rectangle” suffered from excessive conformational heterogeneity after freezing, could the authors comment on the reason for this phenomenon. Does it represent the real status of such single layer DNA origami in solution without freezing?

3. Line 168, the authors claimed that “breathing” motions of DNA helical lattice is striking. However, such phenomenon has been observed before by many works, such as J. Am. Chem. Soc. 2013, 135, 33, 12172-12175; Nano Lett. 2016, 16, 8, 4871-4879.

4. Line 257, I am confused if the RMSD unit is correct. Also, the RMSD value of Pointer V2 is less than 10.

5. Figure 2a-d, detailed explanations for the angle symbol should be provided in the figure caption.

Minor comments:

1. Check the subscript of chemical formulas such as MgCl₂, Line 308, 312.

2. Keep consistence of scientific numbers, e.g., Line 389, 413, 427 and 429.

3. Typos, e.g., Line 199.

Reviewer #3:

Remarks to the Author:

In this work, Dietz and colleagues uses cryo-EM together with molecular-dynamics-based analysis to determine DNA origami structures with improved resolution to reveal more structural details. The authors investigated the global twist deformations for a variety of multi-layer DNA structures. Interestingly, many structures with 7-bp crossover spacing in honeycomb design, which ideally matches with B-DNA twist density, displayed global twist deformations and as a consequence they invalidated the geometrical model in their previous studies. Focused refinement methodologies were adapted to tackle the structural heterogeneity among individual objects, which the authors suggest as the key challenge for revealing high-resolution details. With scanning focused refinement after multi-body refinement, their reconstructed models of best local resolution

revealed the molecular details at the periphery such as backbone phosphates and nicks. They further developed a tool to link the cryo-EM maps, atomic models fitted by ENRG-MD and the design diagrams. It is a very comprehensive study about structural determination of a set of DNA origami structures and the resolution is substantially improved compared to earlier studies. This reviewer recommends a publication when some minor concerns are addressed.

1. It is good to see the efforts by authors trying to determine a single-layer origami structure. Besides DNA origami structures with compact helices, the wireframe architecture has also grown in popularity. It would be nice if the authors can test wireframe structures to see if the streamlined methods can result in structural determination of improved resolution for the different type of DNA origami structures.
2. It makes perfect sense that structural details of interior helices can be better solved. However, this reviewer has some doubts about the claim of 3.8Å resolution. If that was indeed the case, the authors would be able to tell bases. This reviewer would suggest to tune down the claim if such a resolution only applies to very small and specific domains.
3. Using DNA origami as templates to assist structural determination of guest proteins is a goal many researchers in the field are aiming at. It would be wonderful if the authors can provide some examples. Even if it is out of the scope of this study, it would be nice to provide some discussion on the important prospect.

Reviewer #4:

Remarks to the Author:

Kube et al reported a staggering number of 27 cryo-EM structures of different megadalton-scale DNA origami variants and conditions. The supplementary materials show beautiful micrographs with outstanding individual particles, however, the 2D classes, FSC and 3D reconstructions demonstrate flexibility for each of these complexes. The 3D reconstructions could seldom be determined at a sub-nm resolution and varied mostly between 1 and 2nm because of structural fluctuations.

The authors rationally design custom shapes of DNA molecules, and wish to validate their designs experimentally such that they can refine their designs with base-pair interventions. This requires a certain level of detail within their experimental (cryo-EM) map which is not always achievable due to the structural fluctuations mentioned above. To overcome this, they developed a molecular-dynamics-based method for building pseudo-atomic models in a semi-automated way.

The manuscript introduces DNA nanotechnology, explains the need of experimental validation, describes a library of multi-layer DNA objects, provides a guide to global twist deformations, discusses cross-overs and nicks, notes the effect of UV radiation, describes "breathing" motions, presents ways to deal with these including focused classification, multi-body refinement and a new "focal scanning" approach, presents a way to fit initial models into EM densities, and introduces a novel viewer tool. Their maps comprises ~100.000 base pair coordinates (more than what's currently available within the entire PDB) which should "help advance DNA nanotechnology".

With so many items combined in one paper, one might get lost on what the focus was. According to the title and abstract, the main focus is to get an accurate structural validation of designed DNA objects in solution, which was hampered by structural fluctuations. The origin of these structural fluctuations partly lies in the design of the different DNA objects. However, they might also relate to the temperature at which the grids were prepared (22degrees) as well as the composition and pH of the folding buffer (1 mM EDTA, 5 mM TrisBase, 5 mM NaCl; 5 mM MgCl₂; pH 8). The manuscript should at least include a discussion about these aspects. Maybe one sample at three different temps would have provided more systematic (& thermodynamic) insight in structural fluctuations than 3 different samples at one temperature?

Figure 3 shows a nice visualization of the domain motions in one DNA object. The large megadalton DNA objects themselves seems so clear in the micrographs, that this might open ways to better relate the visualized domain movements with (the 2d classes of) the individual particles. Can one give statistics or predicted #percentage of particles for a certain 2d class, and compare those to the real data?

Finally, some words about the style. The manuscript lacks any headings and subheadings. With so many structures and subjects to cover, it reads almost like a novel, taking the reader along the authors adventures. The authors use "we" very frequently and try to grab the readers attention with statements such as "popular route" (line 30), "we address the important problem" (line 19 & 39), "as we will discuss" (line 46), "as we will show" (line 47), "will enable the field to move toward more advanced functionalities" (line 48), "Fortunately, the conclusions drawn in the previous work are not affected" (line 119), "which now does correspond to the previously (wrongly) .. model" (line 124), "Importantly" (lines 139 & 223), "we wondered" (line 150), "If this were true" (line 152), "We now turn from .. to" (line 183), "Frankenstein map" (line 213), "imagine moving a square waffle iron laterally over another square waffle iron" (line 255). This referee would have preferred a clearer structured manuscript with fewer topics, presented straight to the point.

REPLY TO REVIEWERS

We thank the reviewers for their time and effort in critically reviewing our manuscript and for providing their comments. Addressing the comments has helped us improve the manuscript. Please see below a point-to-point response to the comments made by the four reviewers.

Reviewer 1

The authors present a very comprehensive and impactful study that elucidates several important structural features of brick-like DNA origami objects using 3D cryoEM, with rigorous, systematic experimentation, and innovative structural analysis and modeling that should prove very valuable to the nanotechnology field.

Specifically, they first examine how global twist in square lattice objects changes with cross-sectional size, and how it can be removed by altering crossover positions. Interestingly, they also observe global twist deformations for ‘nearly all’ honeycomb objects, which is unexpected given the canonical crossover spacings that match the natural twist of B-form DNA. They also apply their twist correction approach to a previously published FRET ruler. They test the hypothesis that tight/rigid crossovers induce this global twist, by introducing additional bases that relax the crossovers without effect, which demonstrates that this is not the origin. They then test nick placement too and demonstrate that this (random versus planar positioning) does not explain the origin of the global twist, or impact its relaxation. They then apply UV light exposure and find that this does eliminate the global twist. They go on to apply focused refinement to obtain impressive increased resolution in “rigid” subdomains, applied to a tower. Using a focal scanning approach they then are able to increase resolution to 3.8 Angstroms, which for the first time in the origami field offers true base-pair-level structural details to be resolved, which is of clear importance to application to structural as well as functional studies. Finally, they use an atomic modeling based approach to generate best fitting models efficiently, which should prove additionally very helpful for the interpretation of such datasets, and also provide a viewer for interactive viewing of structural data.

A: We thank the reviewer for the appreciation of our given work.

Given the numerous important and impactful contributions of this study, I recommend publication after addressing the several minor, yet also important, points below:

Given that the Rothmund rectangle is one of, if not the, most widely used DNA origami objects in the DNA nanotechnology field for a wide range of applications, both on surfaces and in solution, and it is commonly assumed to be flat, the finding that the Rothmund rectangle may in fact exist in highly warped and heterogeneous sub-states is of major significance; This result would ideally be more prominently displayed in the main figures, or

at least mentioned as an important finding, expanding on its important consequences on applications of this object in nanotechnology, rather than only mentioned in passing and shown in Supplement

A: Thank you for pointing out the importance of this finding. As suggested, we placed these findings more prominently in the main text.

L68: The micrographs showed that the original, uncorrected tile with a crossover density corresponding to a twist density of 10.66 base pairs per turn assumes wrapped-up-like shapes in solution. The high degree of flexibility and the wrapped-up shape are in accordance with previous findings using simulations, on-support atomic force microscopy (AFM) and negative stain electron microscopy (EM), and in solution small-angle X-ray scattering (SAXS) data (17-20) and should be taken into account for in-solution applications.

In the first, square lattice set of designs, it is noted that global twist is observed, but it is not specified whether it is right- or left-handed;

A: Everything has right-handed twist, except for the 6hb with insertions. We clarified the manuscript.

We clarified the twist direction for honeycomb and square lattice structures the first time we mention it in the manuscript.

on L82, it is then stated that global twist was eliminated, but it is not stated how; was this using modeling such as CanDo, or expert intuition, or another approach? While twist modification has previously been performed by the Dietz lab, and others, on numerous occasions, it would still be helpful to specify the approach to the reader here

A: We used expert intuition combined with an approach of iterative design with cryo EM feedback (also from other projects). Figure 2e summarizes our findings which might provide a guideline for twist-correction in the future.

Added to the manuscript:

L112: by reducing the average bases between crossovers to achieve the native 10.5 base pairs per turn.

Similarly, for the second set of honeycomb designs, the results on global twist are very interesting, but again the directionality of the twist appears to be missing, and it is unclear on L109 how the 'refined variant' that is designed to be straight was designed/manipulated, which would be helpful for the reader and future designers to understand

A: Everything has right-handed twist, except for the 6hb with the insertions. We used iterative design with titration of skips and cryo-EM feedback.

Added to the manuscript:

L142: every 21 bases

While nick placement is one possibility for eliminating twist, this assumes that nicks “release” torsional stress that’s propagated along the duplex, which is not necessarily the case; Frank-Kamenetskii has shown that DNA basepairs still stack with significant free energy when nicks are present, and that “gaps” (i.e., deletions of bases) are required to eliminate this stacking; Kim et al., NAR 40: 2862 (2012) and Pan et al., 45: 6284 NAR (2017) examined this effect computationally in CanDo;

A: Other previous work [Lee et al., <https://doi.org/10.1093/nar/gky1189>] indicated the influence of nicks on the global twist of DNA origami structures. We therefore sought to test this hypothesis on our larger crosssection structures. We thus compared designs with aligned and unaligned nick sites and did not observe significant changes in the overall twist of the structure, in accordance with previous findings that the reviewer mentions [Kim et al., NAR 40: 2862 (2012); Pan, Keyao, et al., " Nature communications 5.1 (2014): 1-7; Pan et al., 45: 6284 NAR (2017)].

While the present authors do not appear to have tested this possibility directly by introducing “gaps” (which were also used in Wang et al., JACS 138: 7733 (2016)), their UV results support this “gap” hypothesis, since base deletion including strand rupture is likely to occur, each of which would eliminate the propagation of torsional stress along the duplex;

A: For clarification, we note that UV illumination at the conditions and wavelength used in our work causes mainly pyrimidine crosslinking (mainly T-T and T-C) rather than strand ruptures or base deletions [Taylor, John Stephen. Accounts of chemical research 27.3 (1994): 76-82; Lewis, Roger J., and Philip C. Hanawalt. Nature 298.5872 (1982): 393-396; Rycyna RE, Alderfer JL. Nucleic Acids Res. 1985;13(16):5949-5963.].

While certainly not essential to test this experimentally in the current work, it would appear appropriate to include discussion of gaps along the lines of the above works as an additional useful hypothesis to test

A: As suggested by the reviewer, we added the following sentence to the corresponding discussion in the manuscript:

L214: Previous computational studies (17, 27, 28) indicate that for lattice based DNA origami, the stacking interaction at nick sites might be strong enough to compensate for the missing backbone connection. Introducing gaps might have a more substantial influence on the overall twist compared to nicks.

17. D. N. Kim, F. Kilchherr, H. Dietz, M. Bathe, Quantitative prediction of 3D solution shape and flexibility of nucleic acid nanostructures. *Nucleic acids research* **40**, 2862-2868 (2012).
27. K. Pan *et al.*, Lattice-free prediction of three-dimensional structure of programmed DNA assemblies. *Nature communications* **5**, 5578 (2014).
28. K. Pan, W. P. Bricker, S. Ratanalert, M. Bathe, Structure and conformational dynamics of scaffolded DNA origami nanoparticles. *Nucleic acids research* **45**, 6284-6298 (2017).

Reviewer 2

I think this is a great paper that reveals the structures of DNA origami complexes with nucleotide resolution, using solution cryo-EM and simulation. The reconstructed models with up to 3.8 Å local resolution clearly revealed details such as helical grooves, single-stranded versus double-stranded crossovers, backbone phosphate positions, and single-stranded breaks, along with fast semi-automated pseudo-atomic model building based on MD, this combined method will greatly help the inspections and refinements of various user-defined DNA nanostructures. Importantly, this is very much a step in building structural data bank of DNA nanotechnology, similar to the protein data bank. The work is beautiful! I recommend the acceptance without delay.

A: We thank the reviewer for the kind words on our manuscript.

1. It is better to add a comparison table in SI with previous resolution data to clearly show that current quality of cryo-EM maps is superior to any other DNA origami structure published thus far.

A: As requested, we included a comparison table (Table S1) with previously published reconstructions and their resolutions in the Supplementary Information.

2. Line 52, the authors found that “Rothemund rectangle” suffered from excessive conformational heterogeneity after freezing, could the authors comment on the reason for this phenomenon. Dose it represents the real status of such single layer DNA origami in solution without freezing?

A: Yes, it is generally accepted in the field that the flash-frozen state obtained by freezing rates of $\sim 10,000\text{K/s}$ represents a snapshot of the variety of structures in solution prior to freezing [Dubochet J 1988, Dubochet., 2007, Cheng 2015].

3. Line 168, the authors claimed that “breathing” motions of DNA helical lattice is striking. However, such phenomenon has been observed before by many works, such as J. Am. Chem. Soc. 2013, 135, 33, 12172-12175; Nano Lett. 2016, 16, 8, 4871-4879.

A: We edited the sentence in question and deleted “striking”. It now reads:

L238: Changed to:

Furthermore, we also observed “breathing” motions ...

5. Figure 2a-d, detailed explanations for the angle symbol should be provided in the figure caption.

A: We updated the caption accordingly.

Minor comments:

1. Check the subscript of chemical formulas such as MgCl_2 , Line 308, 312.

2. *Keep consistence of scientific numbers, e.g., Line 389, 413, 427 and 429.*

3. *Typos, e.g., Line 199.*

A: We thank the reviewer for pointing out these minor errors. We reviewed the formulas, corrected typos and the consistency of scientific numbers.

Reviewer 3

In this work, Dietz and colleagues uses cryo-EM together with molecular-dynamics-based analysis to determine DNA origami structures with improved resolution to reveal more structural details. The authors investigated the global twist deformations for a variety of multi-layer DNA structures. Interestingly, many structures with 7-bp crossover spacing in honeycomb design, which ideally matches with B-DNA twist density, displayed global twist deformations and as a consequence they invalidated the geometrical model in their previous studies. Focused refinement methodologies were adapted to tackle the structural heterogeneity among individual objects, which the authors suggest as the key challenge for revealing high-resolution details. With scanning focused refinement after multi-body refinement, their reconstructed models of best local resolution revealed the molecular details at the periphery such as backbone phosphates and nicks. They further developed a tool to link the cryo-EM maps, atomic models fitted by ENRG-MD and the design diagrams. It is a very comprehensive study about structural determination of a set of DNA origami structures and the resolution is substantially improved compared to earlier studies. This reviewer recommends a publication when some minor concerns are addressed.

A: We thank the reviewer for the appreciation.

1. It is good to see the efforts by authors trying to determine a single-layer origami structure. Besides DNA origami structures with compact helices, the wireframe architecture has also grown in popularity. It would be nice if the authors can test wireframe structures to see if the streamlined methods can result in structural determination of improved resolution for the different type of DNA origami structures.

A: We fully agree with the reviewer that wireframe designs are interesting targets for structural analysis. In fact, we have already started collaborations to perform such analysis in the near future. However, we believe adding yet more data with due analysis is out of the scope of the already quite extensive manuscript.

2. It makes perfect sense that structural details of interior helices can be better solved.

A: Results from the “focal scanning refinement” approach presented in this paper suggest that the inferior resolution of the periphery is a consequence of small alignment errors occurring during the reconstruction process when treating the entire object as a rigid body. Small deviations caused by the lattice’ breathing motion amplify, the larger the body and the further away from the alignment center. The “focal scanning refinement” can remove these effects. Therefore, the periphery is not necessarily intrinsically less well-structured than the core of the multi-layer origamis.

However, this reviewer has some doubts about the claim of 3.8Å resolution. If that was indeed the case, the authors would be able to tell bases. This reviewer would suggest to tune down the claim if such a resolution only applies to very small and specific domains.

A: Comparable to beta sheets in proteins which have a spacing of 4.7Å and can be separated at 4Å resolution, for individual bases which are separated by 3.4Å one would need at least achieve this resolution to tell the bases. As requested, we clarified in the manuscript as follows:

L296: Changed to:

Using the focal scanning refinement procedure, we were able to reconstruct fine details of 2x2 parts of the twist tower object with a global resolution of 4.3Å and local resolutions below 4Å (Fig. S33, Movie S4).

3. Using DNA origami as templates to assist structural determination of guest proteins is a goal many researchers in the field are aiming at. It would be wonderful if the authors can provide some examples. Even if it is out of the scope of this study, it would be nice to provide some discussion on the important prospect.

A: As suggested, we added a sentence covering this option to the manuscript:

L51: Likewise, DNA-template-assisted structural determination of proteins (14-16) will benefit from these improvements in cryo-EM methodology for DNA origami.

14. Y. Dong *et al.*, Folding DNA into a Lipid-Conjugated Nanobarrel for Controlled Reconstitution of Membrane Proteins. *Angewandte Chemie* **57**, 2072-2076 (2018).
15. T. G. Martin *et al.*, Design of a molecular support for cryo-EM structure determination. *Proceedings of the National Academy of Sciences of the United States of America* **113**, E7456-E7463 (2016).
16. T. Aksel, Yu, Z., Cheng, Y., Douglas, S.M., Molecular goniometers for single-particle cryo-EM of DNA binding proteins. *bioRxiv*, (2020).

Reviewer 4

Kube et al reported a staggering number of 27 cryo-EM structures of different megadalton-scale DNA origami variants and conditions. The supplementary materials show beautiful micrographs with outstanding individual particles, however, the 2D classes, FSC and 3D reconstructions demonstrate flexibility for each of these complexes. The 3D reconstructions could seldom be determined at a sub-nm resolution and varied mostly between 1 and 2nm because of structural fluctuations.

The authors rationally design custom shapes of DNA molecules and wish to validate their designs experimentally such that they can refine their designs with base-pair interventions. This requires a certain level of detail within their experimental (cryo-EM) map which is not always achievable due to the structural fluctuations mentioned above. To overcome this, they developed a molecular-dynamics-based method for building pseudo-atomic models in a semi-automated way.

The manuscript introduces DNA nanotechnology, explains the need of experimental validation, describes a library of multi-layer DNA objects, provides a guide to global twist deformations, discusses cross-overs and nicks, notes the effect of UV radiation, describes “breathing” motions, presents ways to deal with these including focused classification, multi-body refinement and a new “focal scanning” approach, presents a way to fit initial models into EM densities, and introduces a novel viewer tool. Their maps comprise ~100.000 base pair coordinates (more than what’s currently available within the entire PDB) which should “help advance DNA nanotechnology”.

With so many items combined in one paper, one might get lost on what the focus was. According to the title and abstract, the main focus is to get an accurate structural validation of designed DNA objects in solution, which was hampered by structural fluctuations. The origin of these structural fluctuations partly lies in the design of the different DNA objects. However, they might also relate to the temperature at which the grids were prepared (22degrees) as well as the composition and pH of the folding buffer (1 mM EDTA, 5 mM TrisBase, 5 mM NaCl; 5 mM MgCl₂; pH 8). The manuscript should at least include a discussion about these aspects.

A: As suggested, we have edited the manuscript to also include a discussion on the different factors causing structural fluctuations.

L225: These fluctuations are design-specific and depend on the global shape as well as the topology of the nanostructures. As these motions are driven by the thermal fluctuation of the individual helices, they are presumably dependent on temperature and salt- and buffer-conditions, as well as the overall folding quality of the nanostructure ensemble. Folding defects, caused for example by partially unhybridized or defective oligonucleotides can locally influence the mechanical properties of a helical segment and distort the global shape or act as a hinge for a domain motion.

Maybe one sample at three different temps would have provided more systematic (& thermodynamic) insight in structural fluctuations than 3 different samples at one temperature?

A: The reviewer points to an interesting question, which we plan to address in the future.

Figure 3 shows a nice visualization of the domain motions in one DNA object. The large megadalton DNA objects themselves seems so clear in the micrographs, that this might open ways to better relate the visualized domain movements with (the 2d classes of) the individual particles.

A: Indeed, the domain variability can be seen in class averages and even in individual particles (see image, comparison to 2D projection of multibody maps). However, we think the visualization we present with 3D maps shows the variation in the most clear way.

Figure 1 PCA component 1 of the domain motion. A) Side view of the two extreme conformations of PCA component 1 (orange and blue). The motion is indicated by dashed arrows. B) Bottom view of the conformations shown in A as indicated by the arrow. Dashed arrows represent domain motions. C) Bottom view of the 3D models (1st row), 2D projections of the 3D maps (2nd row), 2D classes representing the orientation of the projections (3rd row), and individual particles of the 2D classes (4th row). The colored boxes indicate the affiliation to the respective maps of A and B.

Can one give statistics or predicted #percentage of particles for a certain 2d class, and compare those to the real data?

A: Please accept our apologies, but we do not understand the point the reviewer is alluding to.

Finally, some words about the style. The manuscript lacks any headings and subheadings. With so many structures and subjects to cover, it reads almost like a novel, taking the reader along the authors adventures. The authors use “we” very frequently and try to grab the readers attention with statements such as “popular route” (line 30), “we address the important problem” (line 19 & 39), “as we will discuss” (line 46), “as we will show” (line 47), “will enable the field to move toward more advanced functionalities” (line 48), “Fortunately, the conclusions drawn in the previous work are not affected” (line 119), “which now does correspond to the previously (wrongly) .. model” (line 124), “Importantly” (lines 139 & 223), “we wondered” (line 150), “If this were true” (line 152), “We now turn from .. to” (line 183), “Frankenstein map” (line 213), “imagine moving a square waffle iron laterally over another square waffle iron” (line 255). This referee would have preferred a clearer structured manuscript with fewer topics, presented straight to the point.

A: We appreciate the input with respect to the style of the paper and tried to adapt the document in order to reflect some of these suggestions. Regarding the frequent use of “we”, we follow the style guide provided by Nature, in particular:

“Nature journals prefer authors to write in the active voice (“we performed the experiment...”) [...]” (<https://www.nature.com/nature-research/for-authors/write>)

As suggested by the reviewer, we include subheading to better organize the diverse discussion induced by large amount of different structures covered in the paper. We hope that these changes help to better communicate the manuscript’s focus to the readers in a more concise fashion.

Changes to the manuscript:

L30: changed to “Programmable self-assembly with DNA is a route to nanofabrication...”

L150: removed “fortunately”

L232: replaced “We now turn from .. to” with subheading

L319: removed “imagine moving a square waffle iron laterally over another square waffle iron”

Reviewers' Comments:

Reviewer #1:

Remarks to the Author:

The authors have addressed my comments so I'm happy to endorse publication of their first-rate work.

Reviewer #2:

Remarks to the Author:

I think this revised paper has been significantly improved in many facets. Hence I think it is acceptable now.

Reviewer #3:

Remarks to the Author:

All the concerns of this reviewer are addressed in the revised manuscript. This reviewer recommends the publication of the manuscript as it is.